# Stable pure-green organic light-emitting diodes toward Rec.2020 standard

Xun Tang [1] ✉, Tuul Tsagaantsooj [1], Tharindu P. B. Rajakaruna[1], Kai Wang [2], Xian-Kai Chen [2], Xiao-Hong Zhang [2], Takuji Hatakeyama [3] & Chihaya Adachi [1,4] ✉

Manipulating dynamic behaviours of charge carriers and excitons in organic light-emitting diodes (OLEDs) is essential to simultaneously achieve high colour purity and superior operational lifetime. In this work, a comprehensive transient electroluminescence investigation reveals that incorporating a thermally activated delayed fluorescence assistant molecule with a deep lowest unoccupied molecular orbital into a bipolar host matrix effectively traps the injected electrons. Meanwhile, the behaviours of hole injection and transport are still dominantly governed by host molecules. Thus, the recombination zone notably shifts toward the interface between the emissive layer (EML) and the electron-transporting layer (ETL). To mitigate the interfacial carrier accumulation and exciton quenching, this bipolar host matrix could serve as a non-barrier functional spacer between EML/ETL, enabling the distribution of recombination zone away from this interface. Consequently, the optimized OLED exhibits a low driving voltage, promising device stability (95% of the initial luminance of 1000 cd m$^{-2}$, $LT_{95}$ > 430 h), and a high Commission Internationale de L'Éclairage y coordinate of 0.69. This indicates that managing the excitons through rational energy level alignment holds the potential for simultaneously satisfying Rec.2020 standard and achieving commercial-level stability.

Realizing a wide colour gamut while maintaining superior operational stability is a critical challenge for state-of-the-art organic light-emitting diodes (OLEDs), which is highly required for the advanced display application[1–5]. In pursuit of accurately reproducing real-world colours, the colour-gamut standard in OLED displays has progressively evolved to the International Telecommunication Union-Radiocommunication Recommendation BT.2020 (Rec.2020), which requires a specified wavelength peak and a narrow full-width-at-half-maximum (FWHM) for primary red-green-blue (RGB) colours[6]. Encouragingly, a highly robust and rigid π-conjugated framework containing alternate boron/nitrogen (B/N) atoms was proposed by Hatakeyama et al. in 2016[7].

Benefitting from the designated localization of electron-withdrawing B atoms and electron-donating N atoms, their multiple-resonance (MR) effect induces a sufficient electronic orbital separation, thereby, triggering the thermally activated delayed fluorescence (TADF) property to harness the triplet excitons[8,9]. Significantly, the suppression of structural deformation at the excited states and molecular vibrations in MR-TADF molecules leads to a narrowband spectrum with excellent colour purity[10–13].

Although the Commission Internationale de L'Éclairage (CIE) x and y coordinates of MR-TADF OLEDs are approaching Rec.2020 standard through diverse molecular designs, the durability of devices remains

[1]Center for Organic Photonics and Electronics Research (OPERA), Kyushu University, 744 Motooka, Nishi-ku, Fukuoka 819-0395, Japan. [2]Institute of Functional Nano & Soft Materials (FUNSOM), Joint International Research Laboratory of Carbon-Based Functional Materials and Devices, Soochow University, Suzhou, Jiangsu 215123, P.R. China. [3]Department of Chemistry, Graduate School of Science, Kyoto University, Kitashirakawa Oiwake-cho, Sakyo-ku, Kyoto 606-8502, Japan. [4]International Institute for Carbon-Neutral Energy Research (I2CNER), Kyushu University, 744 Motooka, Nishi-ku, Fukuoka 819-0395, Japan. ✉e-mail: x-tang@opera.kyushu-u.ac.jp; adachi@cstf.kyushu-u.ac.jp

unsatisfactory[14–16]. This long-term issue is constantly regarded as a primary obstacle in achieving commercial application[17–19]. Besides designing novel MR-TADF emitters incorporating robust molecular conformations, large oscillator strengths (*f*), and fast reverse inter-system crossing (RISC) rates[20–25], recently, to prolong the device operational lifetime, stable phosphors were widely utilized as the effective triplet assistants to sensitize the green and red MR emitters[26–29]. However, like phosphorescent OLEDs, the phosphor-sensitized fluorescence (PSF) system fails to get rid of the noble rare metals, such as iridium (Ir) and platinum (Pt) etc[30,31]. In contrast, pure organic TADF molecules are also promising to be employed as triplet manipulators to stabilize the MR emitters[32,33]. Nevertheless, the advancement in operational stability of MR-OLEDs aided by TADF molecules has significantly lagged[34,35]. This dilemma might be ascribed to the inappropriate TADF molecule selection and the undesired exciton behaviours, including exciton quenching/annihilation and migration under the electrical excitation in the devices[36–38]. Moreover, the use of traditional p-/n-type host materials, like 3,3-di(9H-carbazol-9-yl)biphenyl (mCBP) and 2-[9,9'-spirobi(fluoren)−3-yl]−4,6-diphenyl-1,3,5-triazine (SF3-TRZ) etc., would induce a larger driving voltage operation owing to their wide energy gaps between the adjacent charge carrier transport layers[39–41]. More significantly, their polarity feature makes it difficult to balance the charge carrier injection and transport[42,43]. Thus, the electron/hole mobilities in the emitting layer (EML) of TADF-assistant fluorescence (TAF) devices are highly relied on the doping concentration of TADF assistants[44]. Nevertheless, a low doping concentration of assisted-TADF molecules may not considerably influence the charge carrier behaviours, while a high doping concentration would result in its aggregation-caused quenching, ultimately detrimental to device performance[45]. Therefore, understanding and manipulating charge carrier and exciton behaviours within the EML remains a challenging dilemma.

In this work, the investigation primarily focuses on the manipulation of the recombination zone, as well as the exciton behaviours to sufficiently suppress the detrimental quenching or annihilation processes under electrical excitation, which is aimed at simultaneously achieving the high colour purity and superior stability in OLEDs. In specific, an organoboron-nitrogen-carbonyl compound (*h*-BNCO) with a small FWHM of 38 nm was chosen as the terminal emitter. An electron donor-acceptor (D-A) type molecule, 12-(4,6-diphenyl-1,3,5-triazin-2-yl)−5-phenyl-5,12-dihydroindolo[3,2-a]carbazole (PIC-TRZ2), is used as the host matrix in the devices[46]. Due

to its bipolar characteristic, PIC-TRZ2 possesses a relatively shallow highest occupied molecular orbital (HOMO, −5.6 eV) and a deep lowest unoccupied molecular orbital (LUMO, −2.8 eV), ensuring non-barrier holes or electrons injection from the adjacent carrier transport layers, thereby resulting in a low driving voltage. Meanwhile, a stable green TADF molecule, 1,2,3,5-tetrakis(carbazol-9-yl)−4,6-dicyanobenzene, 2,4,5,6-tetrakis(9H-carbazol-9-yl)isophthalonitrile (4CzIPN), is selected as the assisted dopant to further stabilize the electrically generated triplets[1]. Thus, the 4CzIPN-doped TAF device exhibits promising operational stability, and the lifetime for the luminance degrading to 95% of its initial luminance (LT$_{95}$) was 322 h at the luminance for practical application (1000 cd m$^{-2}$). To further enhance the device stability, the PIC-TRZ2 matrix can additionally serve as a versatile platform to adjust the recombination zone away from the EML/electron transport layer (ETL) interface. This strategy sufficiently extends the LT$_{95}$ to over 400 h, while maintaining a high CIE-y of 0.69.

## Results
### Photophysical properties
As illustrated in Fig. 1a, a bipolar donor-acceptor (D-A)-type molecule, PIC-TRZ2, is utilized as the host material, which would be beneficial for the charge carrier balance within the EML. A traditional green TADF molecule, 4CzIPN, is employed as an assistant to upconvert and stabilize the "dark-state" triplets. *h*-BNCO is selected as the terminal emitter (TE) because of its suitable wavelength (~530 nm), narrowband (38 nm, 0.17 eV), and promising stability[47]. As depicted in Fig. 1b, the radiation transition band from PIC-TRZ2 and 4CzIPN has a sufficient overlap with the ground-state absorption of *h*-BNCO, ensuring the efficient Förster resonant energy transfer (FRET) from PIC-TRZ2 or 4CzIPN to *h*-BNCO. Therefore, it is noticeable that the photo-luminescence (PL) from a 1 wt% *h*-BNCO: PIC-TRZ2 blend film is originated from the emitter *h*-BNCO (Fig. 1c and Supplementary Fig. 1). The PL spectrum possesses a small FWHM of 38 nm (0.17 eV). In contrast, PL from the PIC-TRZ2 neat film is as broad as 123 nm (0.48 eV), and its spectral shape is entirely different from that of the 1 wt% *h*-BNCO: PIC-TRZ2 blend film. Analogously, PL from a 1 wt% *h*-BNCO: 8 wt% 4CzIPN: PIC-TRZ2 ternary film exhibits narrowband emission with an FWHM of 40 nm (0.17 eV). The slight increase in the spectral width could be ascribed to the emission from 4CzIPN.

The transient decay profiles under photoexcitation are further evaluated. In terms of the prompt decay, the prompt decay lifetime of

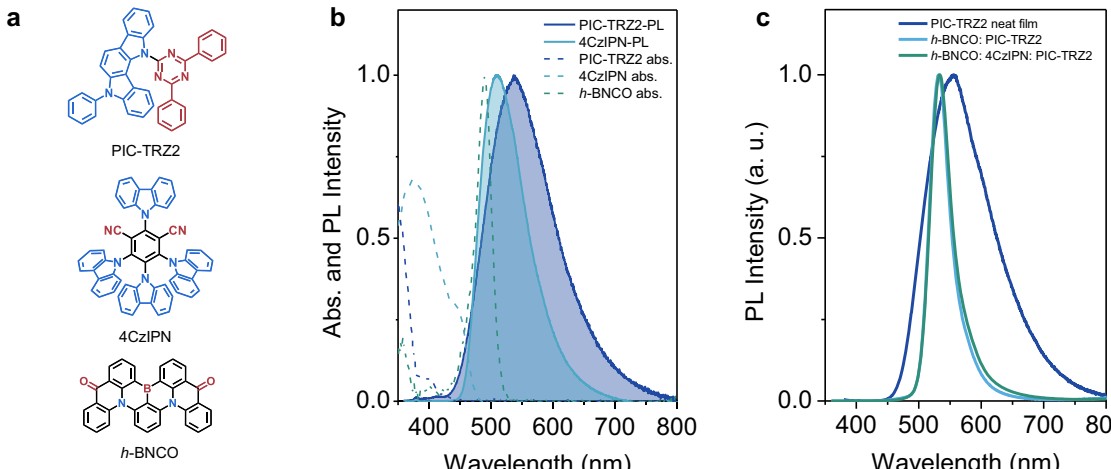

**Fig. 1 | Molecular selection and photophysical properties. a** Molecular structures of PIC-TRZ2, 4CzIPN, and *h*-BNCO. **b** Photophysical properties, including the ground-state absorption and PL of PIC-TRZ2, 4CzIPN, and *h*-BNCO in toluene. **c** PL of the PIC-TRZ2 neat film, 1 wt% *h*-BNCO: PIC-TRZ2, and 1 wt% *h*-BNCO: 8 wt% 4CzIPN: PIC-TRZ2 blend films.

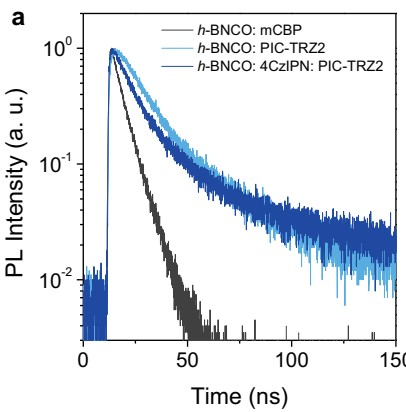

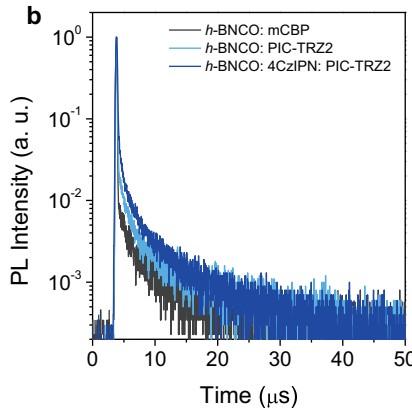

**Fig. 2 | Transient PL decay properties. a** The prompt and **b** the delayed decay properties of 1 wt% *h*-BNCO: mCBP, 1 wt% *h*-BNCO: PIC-TRZ2 and 1 wt% *h*-BNCO: 8 wt% 4CzIPN: PIC-TRZ2 blend films.

the 1 wt% *h*-BNCO: PIC-TRZ2 film ($\tau_p = 17.6$ ns) is much longer than that of the mCBP-based film ($\tau_p = 6.5$ ns) in Fig. 2a. As depicted in Eq. 1,

$$k_{ET} \propto \frac{1}{\tau}, \tag{1}$$

$k_{ET}$ is the energy transfer rate, which is proportional to the fluorescence lifetime of the donors (hosts) in the absence of the acceptors (emitters) ($\tau$)[46]. Considering the comparable spectral overlap integral between PIC-TRZ2/4CzIPN and the emitter, the prolonged radiative process is attributed to its large fluorescence lifetime ($\tau_p = 99$ ns) of PIC-TRZ2 in the absence of *h*-BNCO, which results in a slow FRET rate ($k_{ET}$, around $4.9 \times 10^8$ s$^{-1}$) (Supplementary Figs. 2, 3, and Supplementary Note). By comparison, when excluding the energy transfer process from PIC-TRZ2 to *h*-BNCO by tuning the excitation wavelength to 470 nm, the decay lifetime of 1 wt% *h*-BNCO: PIC-TRZ2 film is 6.5 ns, which is identical to that in the mCBP-based film (Supplementary Figs. 4 and 5). However, because of a small radiative rate ($k_F = 1.1 \times 10^6$ s$^{-1}$) of PIC-TRZ2, which is two orders of magnitude smaller than its $k_{ET}$, the excitons generated at PIC-TRZ2 can also be sufficiently transferred to *h*-BNCO emitters. Benefitting from a shorter $\tau_p$ (17.0 ns) of 4CzIPN, $\tau_p$ of the 1 wt% *h*-BNCO: 8 wt% 4CzIPN: PIC-TRZ2 ternary film decreases to 15.8 ns, indicating a faster $k_{ET}$ from 4CzIPN to the emitters (Fig. 2a). It is worth noting that the incident photons are dominantly absorbed by PIC-TRZ2 host molecules, because the absorption band of PIC-TRZ2 and 4CzIPN is highly overlapped. Owing to the TADF features of PIC-TRZ2 and 4CzIPN, multiple reverse intersystem crossing (RISC) channels appears in the delayed emissive component in Fig. 2b and Supplementary Fig. 5. As shown in Supplementary Fig. 5 and Table S1, taking advantage of their fast RISC rates (around $1.0 \times 10^6$ s$^{-1}$ for PIC-TRZ2 and 4CzIPN)[48], both 1 wt% *h*-BNCO: PIC-TRZ2 and 1 wt% *h*-BNCO: 8 wt% 4CzIPN: PIC-TRZ2 blend films exhibit an accelerated RISC rate of $5.3 \times 10^5$ s$^{-1}$ and $7.4 \times 10^5$ s$^{-1}$, respectively, which is faster than that ($2.9 \times 10^5$ s$^{-1}$) in the mCBP host matrix. The photophysical data are summarized in Tables S1 and S2.

### Electroluminescent performance

To evaluate the electroluminescent (EL) characteristics based on these different emissive layers, three OLEDs with different emissive layers (EMLs) are designed and fabricated as the following architecture in Supplementary Fig. 7: indium tin oxide (ITO), 1,4,5,8,9,11-hexaazatriphenylene hexacarbonitrile (HAT-CN, 10 nm), 9-phenyl-3,6-bis(9-phenyl-9H-carbazol-3-yl)−9H-carbazole (Tris-PCz, 30 nm), mCBP (5 nm), EML (30 nm), SF3-TRZ (10 nm), 30 wt% 8-quinolinolato lithium (Liq): SF3-TRZ (30 nm), Liq (2 nm), and aluminium (Al, 100 nm), where

the EMLs correspond to 1 wt% *h*-BNCO: mCBP (D1), 1 wt% *h*-BNCO: PIC-TRZ2 (D2), and 1 wt% *h*-BNCO: 8 wt% 4CzIPN: PIC-TRZ2 (D3), respectively. As shown in Fig. 3a, narrowband pure-green EL spectra from D1, D2, and D3 are obtained with a small FWHM around 39 nm, indicating the complete energy transfer to *h*-BNCO. In addition, the wavelength peaks ($\lambda_{peak}$) of D1, D2, and D3 are 525, 529, and 530 nm, respectively. Thus, CIE-y values of these three devices are 0.71, 0.70, and 0.68, respectively. In terms of the external quantum efficiencies (EQEs), both D1 and D2 could obtain a high external quantum efficiency (EQE) of around 22% at the low luminance of 10 cd m$^{-2}$. Such a sufficient light outcoupling efficiency can be attributed to the horizontal dipole orientation of *h*-BNCO[47]. Nevertheless, the efficiency rolloff at 1000 cd m$^{-2}$ of D1 is 57.4%, which is significantly larger than that of D2 (26.1%). As a result, D2 exhibits promising operational stability, and LT$_{95}$ is equal to 124 h at the brightness for practical application (1000 cd m$^{-2}$), indicating much better stability than that of D1 (LT$_{95} = 46$ h). To elucidate the underlying reasons for the varied device performances, a deep-red phosphorescent emitter, bis[1-(9,9-dimethyl-9H-fluoren-2-yl)-isoquinoline](acetylacetonate)iridium (III) [Ir(fliq)$_2$(acac)], is employed as an exciton indicator to experimentally determine the recombination zone[49,50]. Ir(fliq)$_2$(acac) is chosen due to its smaller energy gap, allowing the effective trapping of the excitons. Moreover, its red-shifted emission peak ($\lambda_{peak} = 666$ nm) would be easier to differentiate from the emission originating from *h*-BNCO. As shown in Supplementary Fig. 8, a 0.3 nm thickness of Ir(fliq)$_2$(acac) is respectively inserted at the positions of 5, 15, and 25 nm in the EML. It is found that the recombination zone of D1 is mainly localized at the EML/SF3-TRZ interface (Supplementary Fig. 9). In contrast, due to the bipolar characteristics of the PIC-TRZ2, the recombination zone of D2 distributed at the centre of the EML (Supplementary Figs. 10 and 11). To further figure out the degradation process, the change of EL spectral shapes during the continuous operation is analysed in Fig. 3d–f. In the case of D1, an additional spectral peak at 625 nm, which is raised from the SF3-TRZ electromer, gradually appears after operating for 25 h[51]. This distinct shoulder peak indicates the shifting of the recombination zone to the SF3-TRZ3 side and accelerates its degradation. In contrast, a non-desirable spectrum from SF3-TRZ cannot be detected after operating 100 h in D2. Hence, the reduced efficiency rolloff and the improved stability observed in D2 can be partially explained by the reorganization and localization of the recombination zone. In terms of D3, though its maximum EQE is 16.3% due to its lower photoluminescence quantum yield (PLQY ~ 65%) of the EML (Table S2), EQE remains uncompromised at 15.7%, even at the luminance of 1000 cd m$^{-2}$. Significantly, benefiting from the faster RISC rate from 4CzIPN, the operational lifetime (LT$_{95}$) of D3 is substantially extended to 322 h. Noticeably, under continuous electrical excitation, D3

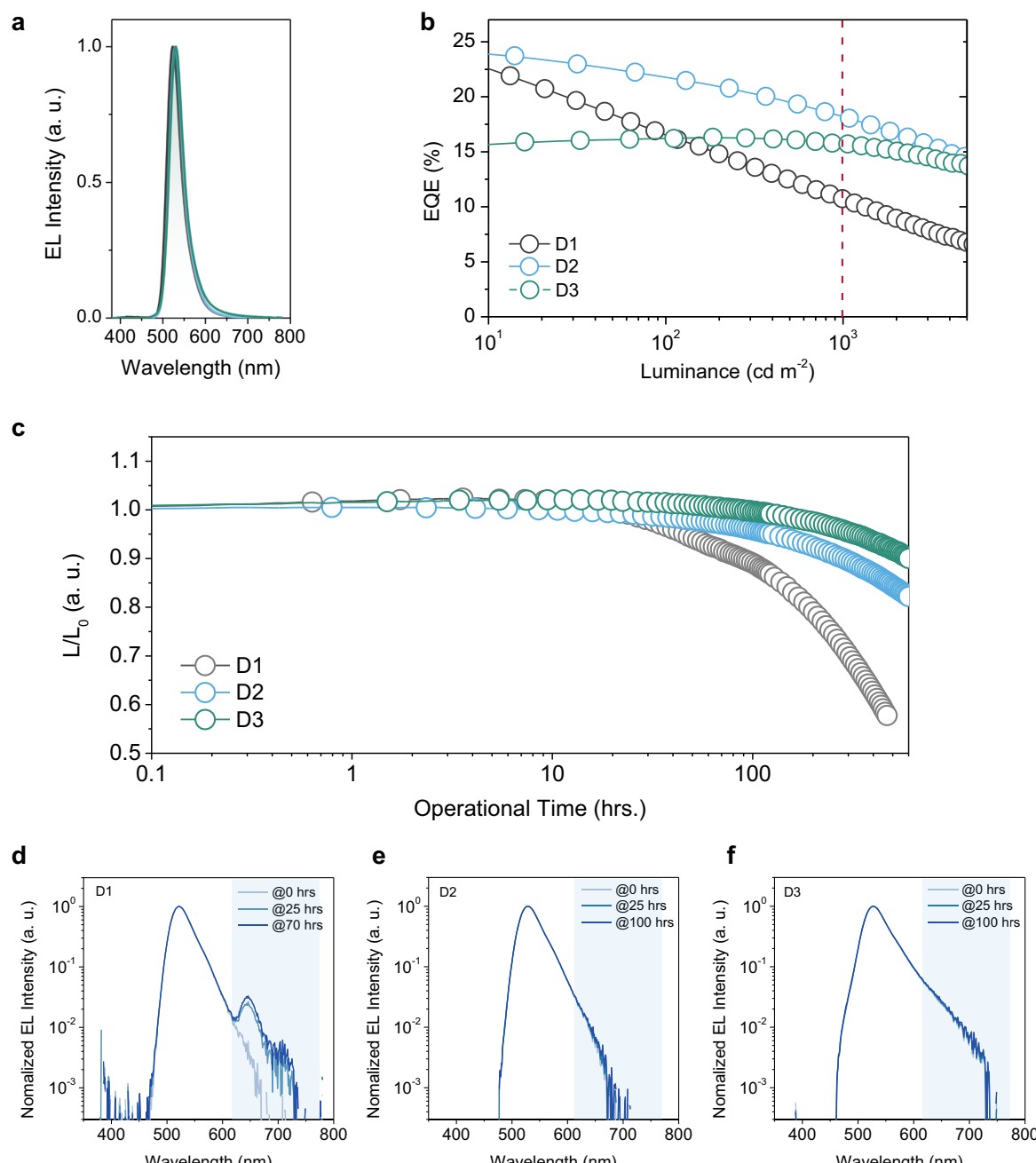

**Fig. 3 | OLED performance and EL spectra during degradation. a** EL spectra, **b** EQEs versus luminance curve, and **c** the operational device stability of D1, D2, and D3, the initial luminance is 1000 cd m$^{-2}$. The normalized EL spectral shapes of **d** D1, **e** D2, and **f** D3 at various operational lifetimes during the degradation.

exhibits very limited shoulder spectral signal from SF3-TRZ, implying minimal exciton diffusion to the SF3-TRZ layer.

**Charge carrier and exciton dynamics**

The dynamics of excitons in D3 is further examined by the Ir(fliq)$_2$(acac) sensing method (Fig. 4a). Interestingly, as influenced by doping 8 wt% 4CzIPN in the EML, the recombination zone of D3 undergoes a remarkable shift, primarily localising around the interface of EML/SF3-TRZ (Fig. 4b and Supplementary Fig. 12). The substantial shift in the recombination zone is strongly relevant to the dynamic behaviours of charge carriers. Consequently, hole-only and electron-only devices (HOD and EOD) based on D2 and D3 are subsequently fabricated (details are provided in Methods section). As illustrated in Fig. 4c, the electron mobility in D2 is comparable to that of holes, suggesting a well-balanced carrier injection and transporting

behaviour within D2. This result is consistent with the distribution of the recombination zone in D2 (Supplementary Fig. 11). In contrast, in D3, there is a remarkable decrease in electron mobility (Fig. 4d), implying that 4CzIPN dopants trap the injected electrons, hindering their transport. It ought to be highlighted that the hole mobility of D3 is identical to that of D2 (Supplementary Fig. S13), indicating holes transport through PIC-TRZ2 because of its shallower HOMO level (−5.6 eV). Consequently, such an imbalance in hole/electron mobility in D3 would make the excitons generate at the EML/SF3-TRZ interface and reduce the EQE.

To gain deep insight into the remarkable difference between D1, D2, and D3, transient EL decay properties are investigated. As depicted in Fig. 4e, upon switching on the voltage pulse, the transient EL response from D1 (black line) is more intense than that of D2 (sky-blue line) within the initial 100 μs. In contrast, 4CzIPN-doped D3 (green line)

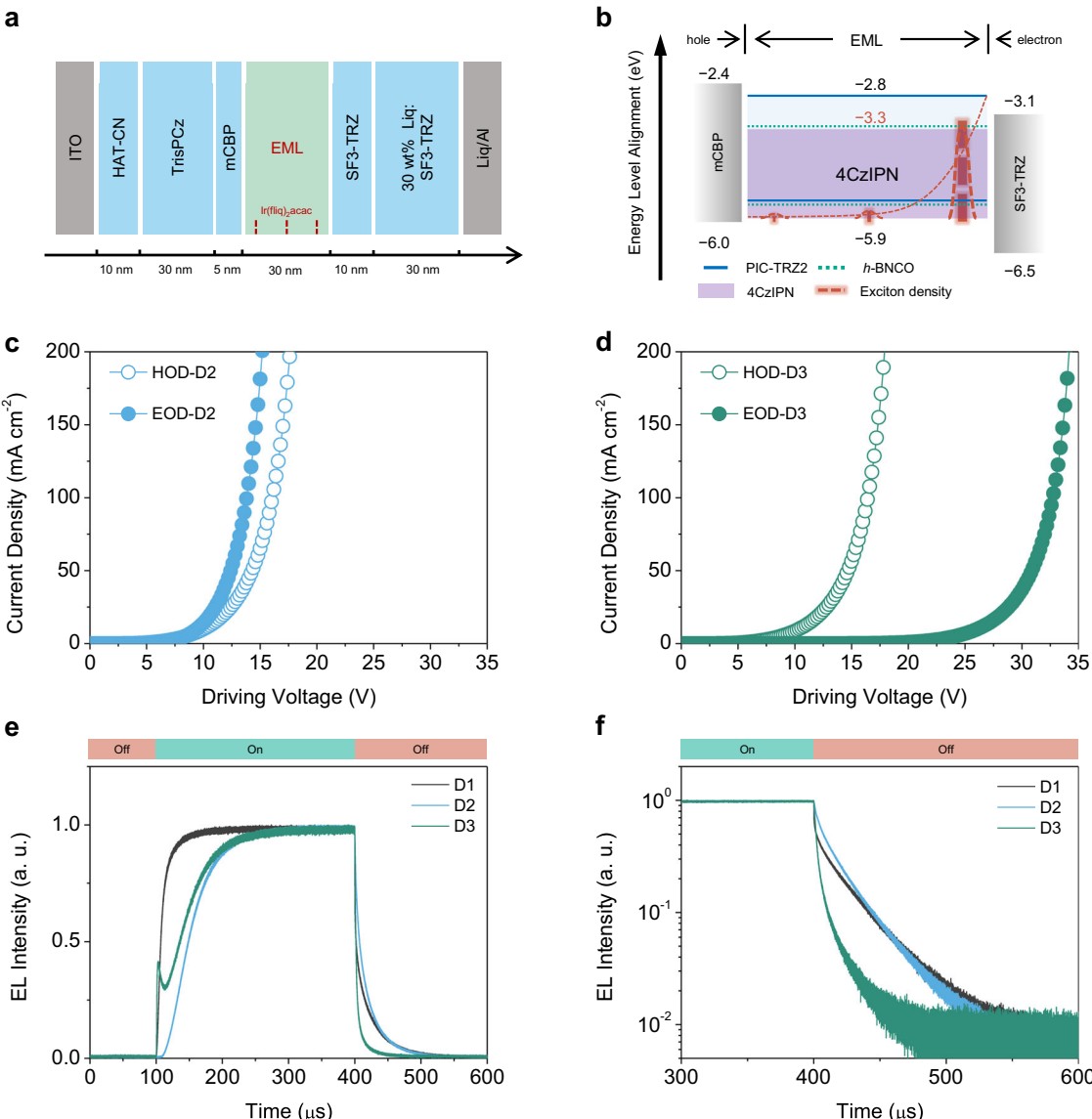

**Fig. 4 | Charge carrier and exciton dynamics in OLEDs. a** OLED structure of D3 with the thickness of each functional layer. **b** Energy level alignment and recombination zone distribution of D3. Hole-only device (HOD) and electron-only device (EOD) of the emitter layers in **c** D2 and **d** D3. **e** Transient EL properties of D1, D2, and D3 (the pulsed width was 300 μs). **f** Transient EL decay properties of D1, D2, and D3 upon switching off the driving voltage.

exhibits a unique EL response. At the very initial time, an abrupt rise in the EL response upon electrical excitation is observed, even surpassing the response from D1 in terms of the slope. Subsequently, there is a noticeable reduction in EL intensity, followed by a gradual increase in EL intensity resembling the trend observed in D2. This abnormal response might be originated from the 4CzIPN-doping effect, since the HOMO and the lowest unoccupied molecular orbital (LUMO) play a crucial role in charge carrier injection and transport under electrical excitation. Specifically, the HOMOs of PIC-TRZ2, *h*-BNCO, and 4CzIPN are measured as −5.6, −5.6, and −5.9 eV, respectively (Supplementary Fig. 14). Additionally, the LUMOs are determined to be −2.8, −3.2, and −3.3 eV for these compounds, respectively (Supplementary Fig. 15). Notably, 4CzIPN exhibits the deepest LUMO, indicating the injected electrons would prefer to be trapped by 4CzIPN. Moreover, upon switching off the power supply, the delayed behaviour of D3 is notably distinct from those of D1 and D2. As shown in Fig. 4f, D1 displays clear dual-decay components, with a triplet lifetime of around 22.9 μs, and the delayed part dominates, accounting for a high proportion of 0.97. For D2, almost no prompted emission could be detected, and the

triplet lifetime is slightly shorter (21.3 μs) than that of the D1 (22.9 μs) because of a faster RISC process from PIC-TRZ2. Remarkably, 4CzIPN-doped D3 has a much shorter triplet lifetime of 3.42 μs. This accelerated RISC process also verifies that the exciton generation and management processes are predominantly influenced by 4CzIPN. Interestingly, the unusual EL reduction of this 4CzIPN-doped ternary film disappears under the photoexcitation (Supplementary Fig. 16), emphasizing the significant impact of the energy level alignment of 4CzIPN on the exciton behaviour under electrical excitation. To further validate the effect of 4CzIPN, a device (D5) based on an 8 wt% 4CzIPN: PIC-TRZ2 EML is fabricated. As shown in Supplementary Figs. 17 and 18, the transient EL profile of D5 exhibits a similar emission response to that of D3. Moreover, the exact recombination zone distribution of D5 is conducted by inserting the Ir(fliq)₂acac indicator, it is found the recombination zone is also localised at EML/SF3-TRZ interface (Supplementary Fig. 19). This suggests that the trapping effect originated from 4CzIPN rather than *h*-BNCO. Additionally, the delayed radiative rate from 4CzIPN in D5 is faster than that from *h*-BNCO in D2, implying energy transfer from 4CzIPN to *h*-BNCO predominates in D3.

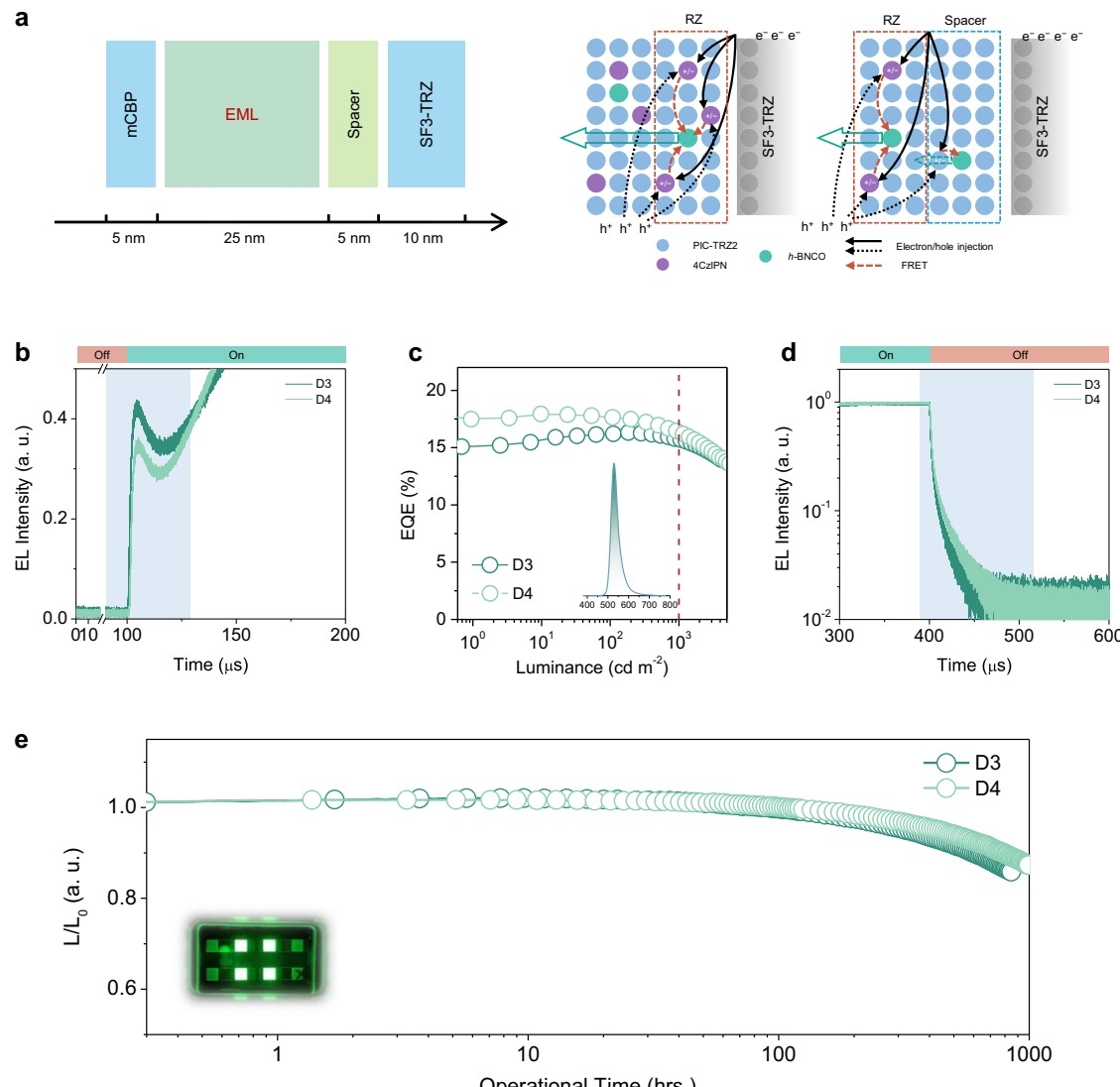

**Fig. 5 | Device performance with the functional layer. a** Illustration of the device structure, carrier transport, exciton generation, energy transfer, and light-emitting process in D3 and D4. **b** Transient EL excitation properties of D3 and D4. **c** EQEs versus luminance curves of D3 and D4. The inset is the EL spectrum of D4 at 1000 cd m$^{-2}$. **d** Transient EL decay properties of D3 and D4 upon switching off the driving voltage. **e** Operational device stability of D3 and D4, the initial luminance is 1000 cd m$^{-2}$. The inset is the operating photograph of D4.

In contrast, an analogue of 4CzIPN, 9,9′,9″,9‴,9⁗-(6-(4,6-diphenyl-1,3,5-triazine-2-yl)benzene-1,2,3,4,5-pentayl)pentakis(9H-carbazole) (5Cz-TRZ), is used as the TADF assistant[52]. As illustrated in Supplementary Fig. 20, 5Cz-TRZ possesses a comparable HOMO of −5.9 eV to that of 4CzIPN, while its LUMO is −3.0 eV, which is shallower than that of 4CzIPN. As depicted in Supplementary Figs. 21 and 22, the recombination zone distribution of 5Cz-TRZ-based device (D6) is determined by the same method, the maximal recombination zone of D6 is close to the hole-transport layer side. Moreover, the transient EL characteristics indicates that D6 (Supplementary Fig. 23) does not exhibit the carrier accumulation or trapping effect that is observed in 4CzIPN-based D3. This finding further underscores the critical role of rational energy level alignment in OLED performance.

## Recombination zone manipulation

To further improve the EL operational stability, suppressing the charge carrier trapping and exciton annihilation is essential. Due to the trapping effect induced by 4CzIPN, the recombination zone dramatically shifts to the EML/SF3-TRZ interface. In Supplementary Figs. 24 and 25, an additional spectral signal from 610 to 700 nm in D3, could respond to the emission from the SF3-TRZ electromer, after operating 300 h, the EL intensity of the SF3-TRZ electromer gradually increases, as well as the deviation of CIE coordinates, suggesting the gradual diffusion of excitons towards the SF3-TRZ side during the device degradation. Hence, suppressing the exciton diffusion to SF3-TRZ would enhance the device stability. The bipolar nature of the PIC-TRZ2 host matrix ensures efficient electron and hole transporting capability. Additionally, the relatively small LUMO energy gap (0.3 eV) between PIC-TRZ2 and SF3-TRZ facilitates efficient electron injection. Therefore, a 5 nm thickness of 1 wt% h-BNCO: PIC-TRZ2 layer is utilized as a functional spacer between the 4CzIPN-doped film and the SF3-TRZ layer (Fig. 5a). In Fig. 5b, the prompt EL response from 4CzIPN⇢h-BNCO (grey-blue background) in D4 is weaker than that of D3, implying the probability of the direct electron trapping by 4CzIPN is reduced because of the functional spacer. According to the analysis of charge carrier mobility in D2, the electron-transport capability in a 1 wt% h-BNCO: PIC-TRZ2 film is slightly stronger than that of hole-transport (Supplementary Fig. 13). Consequently, the exciton recombination maxima would primarily localise at the junction of the spacer/4CzIPN-doped ternary layer, which could be confirmed by the exciton sensing method at the

**Table 1 | Device performance summary of D1-4**

| Device | EQE$_{max}$(%) | EQE @1000 cd m$^{-2}$ (%) | EQE rolloff @1000 cd m$^{-2}$ (%) | LT$_{95}$$^a$ (hrs) | CIE-(x, y)$^b$ |
|---|---|---|---|---|---|
| D1 | 25.1 | 10.7 | 57.4 | 46 | (0.23, 0.71) |
| D2 | 24.9 | 18.1 | 27.3 | 124 | (0.26, 0.70) |
| D3 | 16.3 | 15.7 | 3.7 | 322 | (0.26, 0.68) |
| D4 | 17.9 | 16.6 | 7.3 | 437 | (0.27, 0.69) |

$^a$The initial luminance is 1000 cd m$^{-2}$.
$^b$CIE-(x, y) is measured at 1000 cd m$^{-2}$.

positions of 15, 25, and 30 nm in EML/spacer (Supplementary Fig. 26). Encouragingly, the device performance of D4 is better than D3 (the device performance of D1-D4 is summarized in Table 1). As shown in Fig. 5c, D4 exhibits a slightly higher EQE of 17.9% with a small efficiency rolloff, maintaining an EQE of 16.6% at 1000 cd m$^{-2}$. The slightly improved EQE value might be attributed to the spatial separation betweeb 4CzIPN-doped and SF3-TRZ layers, which enables to suppress the possible exciton quenching or annihilation at the interfaces. Moreover, the transient EL delayed emission lifetime of D4 is 4.96 μs, slightly longer than that (3.42 μs) of D3 (Fig. 5d). This extension in delayed emission, as well as a slightly larger efficiency rolloff of D4 might result from the involvement of the functional spacer (1 wt% h-BNCO: PIC-TRZ2) in the exciton recombination process. Benefitting from the functional spacer, the spike intensity of D4 is slightly weaker than that of D3 when applying the reverse bias voltage (Supplementary Fig. 27). Resultantly, D4 demonstrates an improved operational stability compared to D3, with an LT$_{95}$ extended to 437 h. The much weaker spectral signal from the SF3-TRZ electromer in D4 than that of D3 in Supplementary Fig. 24 indicates the reduced extension of the recombination zone. Comparing with the traditional mCBP-based device (D7), D4 possesses slightly better EL lifetime and much lower driving voltage (Supplementary Figs. 28 and 29). In addition, when further increasing the thickness of the spacer layer to 10, and 20 nm, though their highest EQEs were improved, the EQE rolloff and device lifetime at 1000 cd m$^{-2}$ become worse due to the involvement of spacer layer in exciton recombination process (Supplementary Figs. 30, 31 and Table S3). When replacing the 1 wt% h-BNCO: PIC-TRZ2 spacer by the PIC-TRZ2 neat film and 8 wt% 4CzIPN: PIC-TRZ2 blend film, the corresponding devices exhibit lower EQEs and broader FWHMs (Supplementary Fig. 32). Notably for D4, this superior device stability is achieved in OLEDs made purely of organic elements without rare-metal-containing phosphors (Table S4). Meanwhile, owing to its small FWHM, the corresponding CIE-y of D4 is 0.69. Therefore, the superior device stability, as well as the high colour purity highlight D4 as one of the best pure-green OLEDs aiming for Rec.2020 standard.

## Discussion

The performance of OLEDs, particularly concerning their operational stability, significantly depends on the dynamic distribution of the recombination zone. The comprehension of the inherent degradation mechanisms in the device architecture is crucial to further prolong the EL stability for reaching the commercial level. In our study, it is found that 4CzIPN with a deep LUMO energy level (−3.3 eV) in a bipolar host matrix PIC-TRZ2 sufficiently traps the injected electrons, thus, dramatically changing the carrier dynamic behaviours and drifts the recombination zone to the interface of EML/ETL. Through the systematic transient EL analysis, the accumulation of charge carriers causes the exciton-exciton and exciton-polaron quenching at the fragile interlayer, thereby reducing the EQE values. Although the device lifetime (LT$_{95}$ = 322 h) is much improved due to the assistance of stable 4CzIPN molecules, the unfavourable location of the recombination zone influenced its excellent durability. To further enhance the stability, the host matrix PIC-TRZ2 is employed as a barrier-free spacer between EML/ETL, and feasibly shifting the recombination zone to the

EML side. Consequently, the EQE of the device increases to 17.9% with a small efficiency rolloff and a low driving voltage at 1000 cd m$^{-2}$. More importantly, this strategy effectively improves the device lifetime (LT$_{95}$ = 437 h at 1000 cd m$^{-2}$), incorporating a high CIE-y of 0.69, representing one of the most promising devices toward the Rec.2020 standard.

## Methods

### Materials

All compounds are used as purchased from commercial sources without further purification. h-BNCO is obtained as the synthesis route in the reported reference. The product is purified by column chromatography on a silica gel column with petroleum ether as the eluent.

### Photophysical properties measurement

A spectrophotometer (LAMBDA 950-PKA, PerkinElmer) is used to measure the ground-state ultraviolet-visible (UV-vis) absorption spectra. The steady-state fluorescence at room temperature and phosphorescence at the low temperature (77 K) are recorded by spectrofluorometer (JASCO FP-8600). PLQYs are evaluated by Hamamatsu Photonics C11347-01 Quantaurus-QY. To prevent the interruption from the ambient moisture and oxygen, a 15-min nitrogen-gas bubbling is conducted for the dyes in solution, while the measurement of solid-state films is performed under argon-gas flow. Time-resolved transient photoluminescent (PL) decay profiles are recorded using a Quantaurus-Tau system (C11367-03, Hamamatsu Photonics, Japan) and a time-resolved spectroscopy setup, composed of a third harmonic wave generation, a Nd:YAG/YVO lasing source (EKSPLA PL-2250, the excitation wavelength of 355 nm, and a pulse width of 30 ps), and a streak camera (C10910-01, Hamamatsu Photonics).

### HOMO energy level measurement

The HOMO energy levels of PIC-TRZ2, 4CzIPN, and h-BNCO are achieved through atmospheric ultraviolet photoelectron spectroscopy (Riken Keiki AC-3).

### Device fabrication and characterization

Glass substrates with conductive ITO patterns are ultrasonically cleaned with acetone, isopropanol, and deionized water, then baked in an oven. A 10-min ozone treatment under the exposure of ultraviolet (UV) light before transporting to the glove box is conducted. Organic functional layers and a conductive metallic layer (aluminium, Al) are subsequently deposited onto the pre-patterned ITO substrates in highly vacuumed integrated chambers. This deposition process utilizes different shadow masks for organic and metallic patterns to create a sandwiched-like device architecture. Following the functional layer deposition, the devices are encapsulated by a glass lid with UV-curing epoxy resin in a nitrogen-filled glove box. The device performance is further assessed after encapsulation. Current-voltage-luminance (J-V-L) characteristics are conducted using a calibrated luminance metre (CS-2000, Konica Minolta, Japan) integrated with a Keithley 2400 source metre (Keithley Instruments Inc.).

## Hole-only and electron-only devices (HOD and EOD)

The structure of HOD is ITO/ HAT-CN (10 nm)/ Tris-PCz (30 nm)/ mCBP (5 nm)/ EML/ mCBP (5 nm)/ TrisPCz (30 nm)/ HAT-CN (20 nm)/ Al (100 nm). The structure of EOD is ITO/ Liq (2 nm)/ SF3-TRZ: Liq (30 wt %, 30 nm)/ SF3-TRZ (10 nm)/ EML/ SF3-TRZ (10 nm)/ SF3-TRZ: Liq (30 wt%, 30 nm)/ Liq (2 nm)/ Al (100 nm).

## Transient electroluminescence (EL) measurement

A pulsed transient EL system is established. This system was integrated by a high-definition oscilloscope (2.5 GS/s, HDO4054, TELEDYNE LECROY), a high-speed bipolar driving voltage amplifier (HAS 4101, NF Corporation), a 30 MHz multifunctional pulse generator (WF1974, NF Corporation), a resistance, and a photomultiplier tube (PMT, H11461-02, Hamamatsu Photonics) connected with a power supply for photosensor modules (C10709, Hamamatsu Photonics).

## Device operational stability measurement

The OLED operational stability is evaluated using a luminance metre (SR-3AR, TOPCON) to track the change of the corresponding luminance and spectra under constant direct-current (DC) driving. The initial luminance is 1000 cd m$^{-2}$.

## Data availability

The data that support the findings of this study are available in the supplementary material of this article. Source data are provided with this paper.

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

## Acknowledgements

The authors acknowledge Japan Science and Technology Agency (JST) CREST (grant no. JPMJCR22B3), the Japan Society for the Promotion of Science (JSPS) Specially Promoted Research (grant no. 23H05406), and JSPS International Leading Research (ILR) (grant no. 23K20039). X.T. acknowledges support from JSPS KAKENHI (grant no. 22K20536).

## Author contributions

X.T. and C.A. conceived and supervised the project. X.T. designed all the experiments. X.T., T.T., and T.R. performed the solution and steady-state photophysical measurements of the organic compounds and the corresponding films. X.T. designed and fabricated all the devices and characterized the transient EL properties. K.W., and X.-H.Z. provided *h*-BNCO. X.T. and C.A. wrote the manuscript. K.W., X.-K.C., and T.H. commented on the manuscript. All authors discussed the results and commented on the final manuscript.

## Competing interests

The authors declare no competing interests.
