## [Peer Review File · Nature Communications]

Stable pure-green organic light-emitting diodes toward
Rec.2020 standardREVIEWER COMMENTS

Reviewer #1 (Remarks to the Author):

Tang and co-workers have reported a stable pure-green OLED devices using PIC-TRZ2, 4CzIPN and h-BNCO as the host, TADF assistant, and dopant, respectively, achieving a promising lifetime of $LT_{95} > 430$ hours at an initial luminance of 1000 cd m^{-2} and CIEy of 0.69 through rational energy level alignment and insertion of a non-barrier functional spacer between EML/ETL to enable the distribution of recombination zone away from this interface. Additionally, the authors performed a series of transient EL excitation and decay and measurement of the carrier properties in HOD and EOD to find out the recombination zone and dynamic behaviours of excitons, which further supports the validation of such strategy and the high performances of the devices. These results represent one of the most promising devices toward the Rec.2020 standard and provide a deeply insight on the mechanism of device aging, which is satisfactory for its publication on the journal Nature Communication. However, some issues were not well explained by the authors. Therefore, further revisions on this manuscript are required prior to its consideration of publication. Some comments and suggestion are listed below:

1. This manuscript only provide the PLQY of the 4CzIPN-sensitized EML (65%). In order to comprehensive understand the dynamic behaviours of the luminance, the authors should provide the PLQY of other h-BNCO-based and PIC-TRZ2-based films.
2. The calculation methods of Photophysical properties, such as k_{FET} , k_{RISC} and k_{ISC} are unclear in this manuscript. Please provide.
3. The EL intensity of D3, D4, D5 and D6 with ultra-thin Ir(III)acac layers at different positions in the emissive layer (EML) should also be measured to determine the accurate recombination zone of such devices, which could help us analyzing the shifting process with different device structures.
4. In this manuscript, a 5 nm thickness of 1 wt% h-BNCO: PIC-TRZ2 was chosen as a barrier-free spacer between EML/ETL to shift the recombination zone to the EML side and further enhance the device stability. The authors should also manufacture the devices using merely PIC-TRZ2 and 8 wt% 4CzIPN: PIC-TRZ2 as the spacer for comparison.
5. In Table 1, the EQE roll-off of D4 is high than D3 after inserting such spacer between EML/ETL. Please explain.
6. "Benefitting from a shorter τ_D (17.0 ns) of 4CzIPN ...", the τ_D in this paragraph should be changed to τ_p .
7. In general, the EQE_{max} will be improved after adding a TADF assistant with fast k_{RISC} and short τ_D such as 4CzIPN in the emitting layer with host and dopant. However, the EQE_{max} of D3 and D4 are evidently lower than D1 and D2. Is there some other reasons except for the lower PLQY(65%)?

Reviewer #2 (Remarks to the Author):

How to understand the dynamic behaviors of charge carriers and excitons in organic light-emitting diodes (OLEDs) is difficult but sufficiently relevant to achieve high exciton utilization and device stability. In this work, by studying transient electroluminescence and photoluminescence spectra of different host-guest systems, Tang and his collaborators revealed that incorporating a TADF assistant molecule with a deep LUMO into a bipolar host matrix can effectively trap the injected electrons while maintaining the hole injection and transport behaviors by the host. Therefore, the excitons recombination zone notably shifted toward the interface between EML and ETL, which damaged the device stability. Further, the authors add the bipolar host matrix as a non-barrier functional spacer between EML and ETL to successfully avoid the interfacial carrier accumulation and exciton quenching. The study in this work is systematic and can support its conclusion. Since it indicates that managing the excitons through rational energy level alignment is effective, I think this work can be published in Nature Communications after suitable revision.

1. The specific device structure in this manuscript (interlayer thickness) is not provided, please add.
2. In figure 3d and 3f, how to conclude the degradation mechanism in devices D1 and D3 is the same? Because the spectral shape is quite different, there is a shoulder peak that emerged around 470 nm, as well as a peak around 650 nm, which is not apparent in Figure 3f.
3. In Figure 5, as a long-range coupling phenomenon, the excimer complex still has a peak after optimizing the structure (adding a 5 nm functional layer as a barrier). Why don't continue to increase the functional layer? Please explain.
4. Compared to D3, the increase of exciton utilization is limited but the roll-off is more serious for D4 in high luminance. This seems to contradict the device lifetime data of D3 and D4. What's the difference between device lifetime and roll-off ?

Reviewer #3 (Remarks to the Author):

The authors developed a stable pure-green organic light-emitting diodes (OLEDs) without metal complex by using a bipolar host as a spacer between the light-emitting layer (EML) and electron-transporting layer (ETL), as well as using a sensitized EML composed of bipolar host (PIC-TRZ), TADF sensitizer (4CzIPN) and MR-TADF emitter (h-BNCO). The optimized device of D4 achieved excellent device lifetime with $LT_{95}=437$ hrs at 1000 cd m^{-2} and a high CIE-y of 0.69, representing a significant progress in device stability for pure-green OLEDs without metal complex. Therefore, I am glad to recommend this work for publication after minor revision.

1. The D3 and D4 device has much better device stability than that of D1 and D2. But, why their maximum device efficiency is lower than that of D1 and D2? And how to further improve the device efficiency?

2. The device with the EML of mcbp:4CzIPN:h-BNCO should be fabricated to determine the effect of bipolar host on the stability of sensitized devices.
3. Please the author give the device performance including the efficiency and stability for device D6 with 5CzTRZ as the sensitizer, which could help to understand the function of charge trapping for the sensitizer.
4. Normally, an overshoot could be observed in the EL transient when the pulse is off for devices with charge traps. Why the overshoot has not been observed in D3 and D4?

Point-to-point responses to reviewers' comments

Replies to comments made by reviewers

We thank the reviewers for their insightful comments, which helped us to improve the quality of our manuscript. We have fully addressed the comments in the revised manuscript. The revised sentences and words are highlighted to make them easy to find in the manuscript.

Comments and answers (Reviewer #1)

(General comment)

Tang and co-workers have reported a stable pure-green OLED devices using PIC-TRZ2, 4CzIPN and *h*-BNCO as the host, TADF assistant, and dopant, respectively, achieving a promising lifetime of $LT_{95} > 430$ hours at an initial luminance of 1000 cd m⁻² and CIE_y of 0.69 through rational energy level alignment and insertion of a non-barrier functional spacer between EML/ETL to enable the distribution of recombination zone away from this interface. Additionally, the authors performed a series of transient EL excitation and decay and measurement of the carrier properties in HOD and EOD to find out the recombination zone and dynamic behaviours of excitons, which further supports the validation of such strategy and the high performances of the devices. These results represent one of the most promising devices toward the Rec.2020 standard and provide a deeply insight on the mechanism of device aging, which is satisfactory for its publication on the journal Nature Communication. However, some issues were not well explained by the authors. Therefore, further revisions on this manuscript are required prior to its consideration of publication. Some comments and suggestion are listed below:

(Reply)

We sincerely thank Reviewer #1 very much for appreciating our manuscript and providing us with valuable revision comments and suggestions to optimize our manuscript further.

(Comment 1)

This manuscript only provides the PLQY of the 4CzIPN-sensitized EML (65%). In order to comprehensive understand the dynamic behaviours of the luminance, the authors should provide the PLQY of other *h*-BNCO-based and PIC-TRZ2-based films.

(Reply 1)

We sincerely thank the reviewer for this valuable comment. As shown in **Table R1**, we have conducted PLQYs of *h*-BNCO: mCBP, *h*-BNCO: PIC-TRZ2, *h*-BNCO: 4CzIPN: PIC-TRZ2, PIC-TRZ2, 4CzIPN: PIC-TRZ2, and *h*-BNCO: 5CzTRZ: PIC-TRZ2 films.

Table R1 was included in the revised manuscript and Supplementary Information (SI).

Table R1. Summary of PLQYs of different solid-state films

Film	PLQY (%)
h -BNCO: mCBP	79.1
h -BNCO: PIC-TRZ2	77.1
h -BNCO: 4CzIPN: PIC-TRZ2	65.0
PIC-TRZ2	22.4
4CzIPN: PIC-TRZ2	34.9
h -BNCO: 5CzTRZ: PIC-TRZ2	72.5

(Comment 2)

The calculation methods of photophysical properties, such as k_{FET} , k_{RISC} and k_{ISC} are unclear in this manuscript. Please provide.

(Reply 2)

We thank the reviewer for this important comment. We have added the calculation methods of photophysical properties in the SI for a clear presentation. Please check the Supplementary Note in the revised manuscript and SI.

Estimation of photophysical properties. Rate constants, including k_p , k_F , k_{ISC} , and k_{RISC} , were calculated based on the measured quantum yields and transient PL decay lifetime, following the reported methods:¹⁻³

$$k_p = \frac{1}{\tau_p} \quad (\text{Eq. S1})$$

$$k_F = k_p \Phi_p \quad (\text{Eq. S2})$$

$$k_D = \frac{1}{\tau_D} \quad (\text{Eq. S3})$$

$$k_{\text{ISC}} = \frac{k_F}{\Phi_p} - k_F - k_{\text{IC}} \quad (\text{Eq. S4}) \quad (k_{\text{IC}} = \frac{k_F}{\Phi} - k_F) \quad (\text{Eq. S5})$$

$$k_{\text{RISC}} = \frac{k_p k_D}{k_p - k_{\text{ISC}}} \quad (\text{Eq. S6})$$

where Φ_p and Φ_D are the prompt and delayed quantum yields, respectively, τ_p and τ_D are the prompt and delayed transient PL lifetime, respectively. k_p , k_F , k_D , k_{ISC} , and k_{RISC} are rate constants of prompt fluorescence, singlet fluorescence radiation, delayed fluorescence, intersystem crossing, and reverse intersystem crossing, respectively. The rate constants were corrected by using the Excel document of the updated method.³

1 Masui, K.; Nakanotani, H.; Adachi, C. *Org. Electron.* 2013, 14, 2721.

2 Sano, Y., et al., *J. Am. Chem. Soc.*, 2023, 145, 11504.

3 Tsuchiya, Y. et al., *J. Phys. Chem. A* 2021, 125, 8074–8089.

For k_{FET} , we fabricated the mCBP neat film, 1 wt% *h*-BNCO: mCBP blend film, PIC-TRZ2 neat film, and 1 wt% *h*-BNCO: PIC-TRZ2 blend film, respectively. For mCBP-based films, we measured the transient PL decay profile at the wavelength of 370 nm, which corresponds to the mCBP region. For PIC-TRZ2-based films, we measured at 440 nm, which is the PIC-TRZ2 region. According to the Eq. R7:⁴

$$k_{FET} \approx \frac{1}{\tau'_D} - \frac{1}{\tau_D}, \text{ (Eq. R7)}$$

where k_{FET} is FRET rate, τ'_D and τ_D are the host fluorescence lifetimes with and without the emitter. As obtained from **Fig. R1**, τ'_D and τ_D for mCBP were 0.49 and 3.3 ns, for PIC-TRZ2 were 2.0 and 99 ns, respectively.

4 Ghenuche, P. et al., Nano. Lett. 2014, 8, 4707.

Fig. R1: PL spectrum of **a** 1 wt% *h*-BNCO: mCBP and **c** 1 wt% 1 wt% *h*-BNCO: PIC-TRZ2 films. Transient PL decay curve of **b** mCBP neat film and 1 wt% *h*-BNCO: mCBP blend film at 370 nm; **d** PIC-TRZ2 neat film and 1 wt% *h*-BNCO: PIC-TRZ2 blend film at 440 nm.

Table R2. Photophysical properties of *h*-BNCO-based and PIC-TRZ2-based films.

Film	k_F (10^7 s^{-1})	FRET rate (10^8 s^{-1})	Φ_P	Φ_D	k_{ISC} (10^7 s^{-1})	k_{RISC} (10^5 s^{-1})
1 wt% h -BNCO: mCBP	10.3	17.4	0.685	0.106	3.4	2.9
1 wt% h -BNCO: PIC- TRZ2	2.9	4.9	0.478	0.293	2.6	5.3
PICTRZ2 neat film	0.11	/	0.110	0.114	0.66	9.0
1 wt% h -BNCO: 8 wt% 4CzIPN: PIC- TRZ2	2.5	/	0.403	0.247	3.1	7.4

(Comment 3)

The EL intensity of D3, D4, D5 and D6 with ultra-thin Ir(flip)₂acac layers at different positions in the emissive layer (EML) should also be measured to determine

the accurate recombination zone of such devices, which could help us analyze the shifting process with different device structures.

(Reply 3)

We are thankful to the reviewer for this important comment. We designed and fabricated D3, D4, D5, and D6 with inserting ultra-thin Ir(fliq)₂acac layers at different positions in the emitting layer (EML). As shown in **Figs. R2a** and **R3a**, for D3 and D5 doped with 4CzIPN, the recombination zone distribution was similar, which was mainly localized at the EML/SF3-TRZ interface. For D4 with a 5 nm functional spacer, the Ir(fliq)₂acac ultra-thin layers were placed at the positions of 15, 25, and 30 nm, respectively. In **Fig. R2d**, the recombination zone maximum of D4 was localized at the junction between 4CzIPN-doped EML and the spacer interface, and extended to the extra EML region and the spacer layer. In contrast, for D6 doped with 5Cz-TRZ (**Fig. R3d**), the maximal recombination zone was not localized at the EML/SF3-TRZ interface, while close to the hole-transporting layer (HTL) side. These experimental results were consistent with our assumption and the transient EL properties. These results, figures, and descriptions have been added to the revised manuscript and SI.

Fig. R2: **a** Device structure of D3 with Ir(fliq)₂acac films at 5, 15, and 25 nm of the emitting layer (EML). **b** EL spectra of D3 with Ir(fliq)₂acac at different positions in EML under low/high current density. **c** Illustration of the exact recombination zone distribution in D3. **d** Device structure of D4 with Ir(fliq)₂acac films at 15, 25, and 30 nm of the emitting layer (EML). **e** EL spectra of D4 with Ir(fliq)₂acac at different positions in EML under low/high current density. **f** Illustration of the exact recombination zone distribution in D4.

Fig. R3: **a** Device structure of D5 with Ir(fliq)₂acac films at 5, 15, and 25 nm of the emitting layer (EML). **b** EL spectra of D5 with Ir(fliq)₂acac at different positions in EML under low/high current density. **c** Illustration of the exact recombination zone distribution in D5. **d** Device structure of D6 with Ir(fliq)₂acac films at 5, 15, and 25 nm of the emitting layer (EML). **e** EL spectra of D4 with Ir(fliq)₂acac at different positions in EML under low/high current density. **f** Illustration of the exact recombination zone distribution in D6.

(Comment 4)

In this manuscript, a 5 nm thickness of 1 wt% h-BNCO: PIC-TRZ2 was chosen as a barrier-free spacer between EML/ETL to shift the recombination zone to the EML side and further enhance the device stability. The authors should also manufacture the devices using merely PIC-TRZ2 and 8 wt% 4CzIPN: PIC-TRZ2 as the spacer for comparison.

(Reply 4)

We are thankful to the reviewer for this valuable comment. To compare the device with 1 wt% h-BNCO: PIC-TRZ2 spacer (D4), we fabricated two additional devices with a PIC-TRZ2 neat film and 8 wt% 4CzIPN: PIC-TRZ2 spacers (**Figs. R4a, b, and c**), respectively. In **Fig. R4d**, PIC-TRZ2 and 4CzIPN: PIC-TRZ2-based OLEDs exhibited lower EQEs (12.7% and 17.1%) than that of D4 (17.9%). As concluded by **Fig. R2**, the recombination zone of devices with functional spacers would extend to the spacer regime. Therefore, the lower PLQYs of the PIC-TRZ2 neat film and 8 wt% 4CzIPN: PIC-TRZ2 blend film resulted in lower EQEs. Regarding EL spectra in **Figs. R4d, e**, the FWHMs of these two devices were 41 and 66 nm, respectively, which were broader than that of D4 (~38 nm), resulting in a worse colour purity with smaller CIE-y coordinates of 0.67 and 0.60, respectively. Moreover, the operational lifetime (LT₉₅) of PIC-TRZ2-based and 4CzIPN: PIC-TRZ2-based devices were around 118 and 200 hours, respectively, which were worse than that of h-BNCO: PIC-TRZ2-based D4 (**Fig. R5**).

These results, figures, and discussions have been added to the revised manuscript and SI.

Fig. R4: Device structures of OLEDs with **a** *h*-BNCO: PIC-TRZ2 spacer (D4), **b** PIC-TRZ2 spacer, and **c** 4CzIPN: PIC-TRZ2 spacer. **d** EQE versus luminance curve. **e** EL spectra of D4 and the device with PIC-TRZ2 spacer. **f** EL spectra of D4 and the device with 4CzIPN: PIC-TRZ2 spacer.

Fig. R5: Device lifetime of D4, OLEDs with a PIC-TRZ2 spacer, and with a 4CzIPN: PIC-TRZ2 spacer.

(Comment 5)

In Table 1, the EQE roll-off of D4 is higher than D3 after inserting such spacer between EML/ETL. Please explain.

(Reply 5)

We are thankful to the reviewer for this valuable comment. This phenomenon can be explained by the following reasons.

(i) As shown in **Fig. R6**, the functional spacer in D4 would effectively shift the recombination zone to the junction between the 4CzIPN-doped film and the spacer layer. Thus, the suppressed exciton quenching or annihilation at the interface would increase the EQE at the low luminance (**Fig. R6c**); while the efficiency rolloff became larger at

the higher luminance (like at 1000 cd m^{-2}), because the spacer part would be involved in the recombination process when increasing the current density.

(ii) Furthermore, we designed and fabricated devices with thicker spacer layers in **Fig. R7** and **Table R3**. As can be seen, when further increasing the spacer thickness to 10 and 20 nm (D4-1 and D4-2), the recombination zone distributed on the spacer layer, resulting in higher EQEs (16.3, 17.9, 19.4, and 19.8% for D3, D4, D4-1, and D4-2, respectively) at the low luminance (or low current density) region, while a larger EQE rolloff (3.7, 7.3, 12.4, and 13.1% for D3, D4, D4-1, and D4-2, respectively) at the high luminance (1000 cd m^{-2}).

These results, figures, table, and discussions have been added in the revised manuscript and SI.

Fig. R6: Device structure of **a** D3 and **c** D4 with $\text{Ir}(\text{fliq})_2\text{acac}$ at different positions. The recombination zone distribution of **b** D3 and **d** D4.

Fig. R7: **a** Device structures of OLEDs with different spacer thicknesses. **b** EQE versus luminance curve of these OLEDs. **c** Transient EL decay properties of devices with different spacer thicknesses.

Table R3. Summary of device performance with different spacer thicknesses.

Device	Spacer thickness (nm)	EQE_{max} (%)	EQE (%) @ 1000 cd m^{-2}	Rolloff (%)
--------	-----------------------	-------------------------------	---	-------------

D2	0	24.5	18.1	26.1
D3	0	16.3	15.7	3.7
D4	5	17.9	16.6	7.3
D4-1	10	19.4	17.0	12.4
D4-2	20	19.8	17.2	13.1

(Comment 6)

“Benefitting from a shorter τ_d (17.0 ns) of 4CzIPN...”, the τ_d in this paragraph should be changed to τ_p .

(Reply 6)

We are thankful to the reviewer for pointing out this mistake. We have revised τ_d to τ_p accordingly. Also, we have carefully checked and brushed up the manuscript. Please check the highlighted parts in the revised manuscript and SI.

(Comment 7)

In general, the EQE_{max} will be improved after adding a TADF assistant with fast k_{RISC} and short τ_D such as 4CzIPN in the emitting layer with host and dopant. However, the EQE_{max} of D3 and D4 are evidently lower than D1 and D2. Are there some other reasons except for the lower PLQY (65%)?

(Reply 7)

We are thankful to the reviewer for this valuable question. According to the following equation R1 to evaluate the EQE:

$$\eta_{EQE} = \eta_{int} \times \eta_{out} = \gamma \times \chi_{ST} \times \eta_{PL} \times \eta_{out}, \text{ (Eq. R1)}$$

where γ is the factor as an imperfect balance of electrons and holes in the emission zone, χ_{ST} is the spin formation ratio, η_{PL} is the PLQY of the emitting layer, and η_{out} is the outcoupling efficiency. Besides the relatively lower PLQY (η_{PL}) of 65%, the unbalanced holes and electrons in the emission zone (**Fig. R8**) decrease γ , which also resulted in the reduction of the EQEs.

Fig. R8: Hole-only device (HOD) and electron-only device (EOD) of **a** D2 and **b** D3.

Comments and answers (Reviewer #2)

(General comment)

How to understand the dynamic behaviors of charge carriers and excitons in organic light-emitting diodes (OLEDs) is difficult but sufficiently relevant to achieve high exciton utilization and device stability. In this work, by studying transient electroluminescence and photoluminescence spectra of different host-guest systems, Tang and his collaborators revealed that incorporating a TADF assistant molecule with a deep LUMO into a bipolar host matrix can effectively trap the injected electrons while maintaining the hole injection and transport behaviors by the host. Therefore, the excitons recombination zone notably shifted toward the interface between EML and ETL, which damaged the device stability. Further, the authors add the bipolar host matrix as a non-barrier functional spacer between EML and ETL to successfully avoid the interfacial carrier accumulation and exciton quenching. The study in this work is systematic and can support its conclusion. Since it indicates that managing the excitons through rational energy level alignment is effective, I think this work can be published in Nature Communications after suitable revision.

(Reply)

We sincerely thank reviewer #2 for appreciating our work. We also thank the reviewer for his/her careful reading and insightful comments and suggestions, which have strongly helped us to further improve our manuscript.

(Comment 1)

The specific device structure in this manuscript (interlayer thickness) is not provided, please add.

(Reply 1)

We sincerely thank the reviewer for pointing out this detail. We have added and highlighted the specific device structures including the thickness of each functional layer in the revised Fig. 4, Fig. 5, and Supplementary Fig. 7 in the manuscript and SI.

Fig. R9: a, Device structure of D3 with the thickness of each functional layer. b, Energy level alignment and recombination zone distribution of D3.

Fig. R10: **a** Illustration of carrier transport, exciton generation, energy transfer, and light-emitting process in D3 and D4. **b** Transient EL excitation properties of D3 and D4. **c** EQE versus luminance curves of D3 and D4. The inset is the EL spectrum of D4 at 1000 cd m^{-2} . **d** Transient EL decay properties of D3 and D4.

(Comment 2)

In figure 3d and 3f, how to conclude the degradation mechanism in devices D1 and D3 is the same? Because the spectral shape is quite different, there is a shoulder peak that emerged around 470 nm, as well as a peak around 650 nm, which is not apparent in Figure 3f.

(Reply 2)

We sincerely thank the reviewer for the important question. First, we measured the PL stability of the *h*-BNCO: mCBP and *h*-BNCO: 4CzIPN: PIC-TRZ2 films. As shown in **Fig. R11**, both films exhibited comparable PL stability, in specific, LT_{80} for *h*-BNCO: mCBP and *h*-BNCO: 4CzIPN: PIC-TRZ2 films were 18 and 14 hours, respectively. This indicated that molecular photostability is not the main reason for degradation. In general, the recombination zone distribution is the primary factor that affects the device stability. In contrast to the PIC-TRZ2-based D2, the recombination zone of D1 and D3 has been experimentally convinced to be localized at the EML/ETL interface (**Fig. R12**). When further analyzing the EL spectral change, we found that the emission from SF3-TRZ electromer (the shoulder peak at around 650 nm) was more significant at D1 and D3. Meanwhile, its intensity increased as the operational time went on, indicating the recombination zone of both D1 and D3 would gradually extend to the SF3-TRZ layer. Therefore, the deviations of CIE-x and -y coordinates (**Fig. R13**) in D1 and D3 were (0.0131, 0.0128) and (0.0013, 0.0026), respectively, which was much larger than that of D2 (0.0005, 0.0009).

The stronger electromer intensity of D1 than D3 can be partially explained that (i) the larger current density and driving voltage required by D1 to reach 1000 cd m^{-2} , which would promote the recombination zone shift to the SF3-TRZ layer and (ii) the triplet lifetime (τ_d) of D1 was $22.9 \mu\text{s}$, which was much longer than that of D3 ($\tau_d = 3.4 \mu\text{s}$) (**Fig. R14**). The much longer triplet exciton lifetime and a slower RISC process would result in the exciton diffusion and annihilation, resulting in the worse EL lifetime of D1.

These results, including figures, table, and discussions, have been added to the revised manuscript and SI.

Fig. R11: PL stability of *h*-BNCO: mCBP and *h*-BNCO: 4CzIPN: PIC-TRZ2 films under the intensity of 25 mW/cm².

Fig. R12: Experimental recombination zone distribution of **a** D1 and **b** D3. **c** EL spectra of D1 at around the 0, 200, 400, and 600 hours, **d** D3 at around 0, 300, 600, and 800 hours.

Fig. R13: Experimental recombination zone distribution of **a** D1 and **b** D3. **c** EL spectra of D1 at around 0, 200, 400, and 600 hours, **d** D3 at around 0, 300, 600, and 800 hours.

Fig. R14: Transient EL delayed components of D1 and D3.

(Comment 3)

In Figure 5, as a long-range coupling phenomenon, the excimer complex still has a peak after optimizing the structure (adding a 5 nm functional layer as a barrier). Why don't continue to increase the functional layer? Please explain.

(Reply 3)

We thank the reviewer for this important comment. To better understand the effects of the functional layer, we designed and fabricated OLEDs with different spacer thicknesses (0, 5, 10, and 20 nm, **Fig. R15a**). As can be seen in **Fig. R15b**, when increasing the thickness of the functional spacer from 0 to 20 nm, the highest EQEs of devices increased from 16.3% (D3) to approaching 20% (D4-2). However, the efficiency rolloff also becomes larger. It can be explained that the 1 wt% *h*-BNCO: PIC-TRZ2 spacer layer was gradually involved in recombination zone distribution. Then, we conducted the transient EL decay measurement. It can be found from **Fig. R15c** that the abnormal signal intensity decreased with increasing the spacer thickness and finally disappeared when the spacer was 20 nm. This finding further implied that the recombination zone of D4-2 is mainly localized in the functional spacer. Moreover, their device operational stability was different. As shown in **Fig. R16**, D4 with a 5 nm spacer exhibited the best device lifetime, when further increasing the spacer thickness up to 10 and 20 nm, their corresponding device lifetime decreased, specifically, D4-1 and D4-2 showed LT₉₅ of 306 and 203 hours, respectively. The decrease in device lifetime when further increasing spacer thickness can be explained by the recombination zone distribution in the *h*-BNCO: PIC-TRZ2 spacer layer. These results, including figures, table, and discussions, have been added in the revised manuscript and SI.

Fig. R15: **a** Device structures of OLEDs with different spacer thicknesses. **b** EQE versus luminance curve of these OLEDs. **c** Transient EL decay properties of devices with different spacer thicknesses.

Fig. R16: Device lifetime of D2, D3, D4, D4-1, and D4-2.

(Comment 4)

Compared to D3, the increase of exciton utilization is limited but the roll-off is more serious for D4 in high luminance. This seems to contradict the device lifetime data of D3 and D4. What's the difference between device lifetime and roll-off?

(Reply 4)

We sincerely thank the reviewer for the valuable question. First, we would like to explain the rolloff difference between D3 and D4. As can be seen, the major difference in EQEs between D3 and D4 appeared at the low luminance $< 200 \text{ cd m}^{-2}$ (or low current density) region. Specifically, the EQE of D3 slowly increased to the maximum, while D4 had a very stable EQE. This can be explained by the following reasons.

(i) As shown in **Fig. R17**, the functional spacer in D4 would effectively shift the recombination zone to the junction between the 4CzIPN-doped film and the spacer. Thus, the *h*-BNCO: PIC-TRZ2 spacer would be involved in the recombination process, which increases the EQE at the low luminance, while slightly increasing the efficiency rolloff at the higher luminance (like at 1000 cd m^{-2}) (**Table R4**).

(ii) Additionally, the recombination zone was away from the SF3-TRZ interface, which could suppress the exciton shifting and quenching at the SF3-TRZ interface and benefit the device lifetime.

Hence, there was a trade-off between obtaining a smaller efficiency rolloff and manipulating the recombination zone in 4CzIPN-doped EML. It was evident that when further increasing the spacer thickness to 10 or 20 nm, EQEs at low luminance became higher, but efficiency rolloff at 1000 cd m^{-2} also became larger, which leads to a worse device lifetime. D4 obtained a good balance to achieve the best device stability. Thus, there was no contradiction between D3 and D4.

These results, including figures, table, and discussions, have been added to the revised manuscript and SI.

Fig. R17: Device structure of **a** D3 and **c** D4 with $\text{Ir}(\text{fliq})_2\text{acac}$ at different positions. The recombination zone distribution of **b** D3 and **d** D4.

Table R4. Summary of device performance with different spacer thicknesses.

Device	Spacer thickness (nm)	EQE_{max} (%)	EQE (%) @ 1000 cd m^{-2}	Rolloff (%)
D2	0	24.5	18.1	26.1
D3	0	16.3	15.7	3.7
D4	5	17.9	16.6	7.3
D4-1	10	19.4	17.0	12.4
D4-2	20	19.8	17.2	13.1

Comments and answers (Reviewer #3)

(General comment)

The authors developed a stable pure-green organic light-emitting diodes (OLEDs) without metal complex by using a bipolar host as a spacer between the light-emitting layer (EML) and electron-transporting layer (ETL), as well as using a sensitized EML composed of bipolar host (PIC-TRZ), TADF sensitizer (4CzIPN) and MR-TADF emitter (h-BNCO). The optimized device of D4 achieved excellent device lifetime with $LT_{95}=437$ hrs at 1000 cd m^{-2} and a high CIE-y of 0.69, representing a significant progress in device stability for pure-green OLEDs without metal complex. Therefore, I am glad to recommend this work for publication after minor revision.

(Reply)

We sincerely thank reviewer #3 very much for appreciating our manuscript and providing insightful comments and suggestions to further improve our manuscript.

(Comment 1)

The D3 and D4 device has much better device stability than that of D1 and D2. But, why their maximum device efficiency is lower than that of D1 and D2? And how to further improve the device efficiency?

(Reply 1)

We sincerely thank the reviewer for pointing out this question. The maximum EQEs of 4CzIPN-doped hyperfluorescent devices D3 and D4 were lower than those of D1 and D2, because the PLQY of 1 wt% *h*-BNCO: 8 wt% 4CzIPN: PIC-TRZ2 film was 65.0%, which was lower than those of 1 wt% *h*-BNCO: mCBP (79.1%) and 1 wt% *h*-BNCO: PIC-TRZ2 (77.1%) films. One of the strategies to further improve the device EQE is to increase the spacer thickness. As shown in **Fig. R18**, when increasing the spacer thickness to 20 nm, the highest EQE can approach 20%. Another strategy is to change the bipolar host materials or TADF assistant molecules to further improve the PLQYs and EQEs.

These results, including figures, table, and discussions, have been added in the revised manuscript and SI.

Fig. R18: a, Device structures of OLEDs with different spacer thicknesses. b, EQE versus luminance curve of these OLEDs

(Comment 2)

The device with the EML of mCBP: 4CzIPN: *h*-BNCO should be fabricated to determine the effect of bipolar host on the stability of sensitized devices.

(Reply 2)

We thank the reviewer for this valuable comment. As shown in **Fig. R19**, the OLED based on the 1 wt% *h*-BNCO: 8 wt% 4CzIPN: mCBP emitting layer was fabricated. LT₉₅ (measured from 1000 cd m⁻²) of this mCBP-based OLED was around 419 h, which is slightly shorter than that of D4 (LT₉₅=437 h). Notably, in **Figs. R17c** and **d**, the driving voltage of the mCBP-based device is much larger than that of bipolar host-based D3 and D4. In detail, at 1000 cd m⁻², the driving voltage of the mCBP-based device was around 7 V. In contrast, the driving voltage of PIC-TRZ2-based D3 and D4 was around 4.5 V, which was much smaller, indicating the advantage of utilizing the bipolar host.

These results, including figures, table, and discussions, have been added in the revised manuscript and SI.

Fig. R19: OLED performance of mCBP-based hyperfluorescence device D7. **a**, Device structure of D7. **b**, Device operational stability that measured at the luminance of 1000 cd m⁻². The inset is the EL spectrum at 1000 cd m⁻². **c**, Current density versus voltage curve. **d**, Luminance versus voltage curve.

(Comment 3)

Please the author gives the device performance including the efficiency and stability for device D6 with 5CzTRZ as the sensitizer, which could help to understand the function of charge trapping for the sensitizer.

(Reply 3)

We thank the reviewer for this valuable suggestion. As shown in **Fig. R20**, the 5CzTRZ-based device exhibited an EQE of 21%, and the FWHM of the EL spectrum at 1000 cd m⁻² was 39 nm, indicating the complete energy transfer from 5CzTRZ to *h*-BNCO. The device stability was also measured. LT₉₅ at the initial luminance of 1000

cd m⁻² was 71 h. In addition, the dynamic recombination zone distribution of D6 has been investigated by inserting the deep red indicator at 5, 15, and 25 nm in EML. In **Fig. R21**, the recombination zone maximum was localized at the hole-transporting side, indicating electrons would not be trapped by 5Cz-TRZ with a shallower LUMO. These results, including figures, table, and discussions, have been added in the revised manuscript and SI.

Fig. R20: OLED performance of 5CzTRZ-based device D6. **a**, Device structure. **b**, EQE versus luminance curve. The inset is the EL spectrum at 1000 cd m⁻². **c**, Current density-voltage-luminance curve, and **d**, device operational stability.

Fig. R21: Recombination zone distribution of 5CzTRZ-based device D6. **a**, Device structure with deep red emitter Ir(fliq)₂acac at different positions. **b**, EL spectra of OLEDs with Ir(fliq)₂acac at 5, 15, and 25 nm in EML under around 0.01 and 4 mA cm⁻². **c**, Illustration of the recombination zone distribution in D6.

(Comment 4)

Normally, an overshoot could be observed in the EL transient when the pulse is off for devices with charge traps. Why the overshoot has not been observed in D3 and D4?

(Reply 4)

We thank the reviewer for the valuable question. Usually, an overshoot or spike would appear after applying the reverse bias voltage. As shown in **Fig. R22**, when a reverse bias voltage (-10 and -20 V) was applied after turning off the forward voltage pulse, an overshoot can be observed in D3 and D4, indicating the recombination of the trapped charges. Specifically, the spike intensity of D3 is slightly larger than that of D4. The transient EL decay property under reverse bias voltage has been included in the revised manuscript and SI.

Fig. R22: Transient EL decay properties of D3 and D4 under pulsed electrical excitation as dependent on different reverse bias voltage (0, -10 , and -20 V) after turning off the driving voltage.

REVIEWERS' COMMENTS

Reviewer #1 (Remarks to the Author):

The authors have addressed all the issues I concerned and this manuscript can be accepted.

Reviewer #2 (Remarks to the Author):

After revision, the rationality of this manuscript has been greatly improved. Therefore, I am glad to recommend this work for publication. If possible, whether author can try to explain the "the abnormal signal" in Figure R15.

Reviewer #3 (Remarks to the Author):

All the issues have been solved. I suggest the publication of this manuscript.

Point-to-point responses to reviewers' comments

Replies to comments made by reviewers

We thank the reviewers for their insightful comments, which helped us to improve the quality of our manuscript. We have fully addressed the comments in the revised manuscript.

Comments and answers (Reviewer #1)

(General comment)

The authors have addressed all the issues I concerned, and this manuscript can be accepted.

(Reply)

We sincerely appreciate Reviewer #1 for recommending our manuscript for acceptance.

(Comment 1)

Reviewer #1 requests to confirm your claim on the CIE(x,y) coordinates on page 10 of the response to referees letter, in which the deviations of CIE(x,y) of D3 appear to be comparable to (but not much larger than) those in D2 in Figure R13. Reviewer #1 also points out that the figure legend for Figure R13 might be incorrect (as it is the same as Figure R12)

(Reply)

We sincerely thank Reviewer #1 for pointing out this issue. We have modified the presentation of "the CIE-x,y derivation of D3 is larger than D2." Additionally, we revised the legend of Figure R13 (Supplementary Figure 25) to "Supplementary Figure 25: The deviation of the CIE-x coordinate (a) and the CIE-y coordinate in D1-3 (b)."

Comments and answers (Reviewer #2)

(General comment)

After revision, the rationality of this manuscript has been greatly improved. Therefore, I am glad to recommend this work for publication. If possible, whether author can try to explain the "the abnormal signal" in Figure R15.

(Reply)

We sincerely appreciate Reviewer #2 for recommending our manuscript for acceptance. The abnormal signal in Figure R15 could be partially explained by the exciton and polaron annihilation. According to the analysis in the main text, 4CzIPN can effectively trap the electrons. Thus, electron accumulation would easily occur at the

4CzIPN-doped film and SF3-TRZ heterostructure, resulting in the exciton-polaron annihilation, as well as the EL intensity decline. Therefore, when inserting the spacer and increasing its thickness, the electron accumulation at the interface would be reduced. The related investigation will be further conducted in future work.

Comments and answers (Reviewer #3)

(General comment)

All the issues have been solved. I suggest the publication of this manuscript.

(Reply)

We sincerely appreciate Reviewer #3 for recommending our manuscript for acceptance.

(Comment 1)

Reviewer #3 suggests to clearly mark that the EQE roll-off is at 1000 cd/m² in Table 1 in the main text.

(Reply 1)

We appreciate Reviewer #3 for pointing out this detail, we have added “1000 cd m⁻²” in Table 1 in the main text.